# The Detection of Foodborne Pathogenic Bacteria in Seafood Using a Multiplex Polymerase Chain Reaction System

**DOI:** 10.3390/foods11233909

**Published:** 2022-12-04

**Authors:** Pengzhen Li, Xiaoxuan Feng, Baiyan Chen, Xiaoying Wang, Zuyue Liang, Li Wang

**Affiliations:** Guangdong Provincial Key Laboratory of Food Quality and Safety, College of Food Science, South China Agricultural University, Guangzhou 510642, China

**Keywords:** seafood, foodborne pathogenic bacteria, multiplex PCR detection, food safety

## Abstract

Multiplex polymerase chain reaction (PCR) assays are mainly used to simultaneously detect or identify multiple pathogenic microorganisms. To achieve high specificity for detecting foodborne pathogenic bacteria, specific primers need to be designed for the target strains. In this study, we designed and achieved a multiplex PCR system for detecting eight foodborne pathogenic bacteria using specific genes: *toxS* for *Vibrio parahaemolyticus*, *virR* for *Listeria monocytogenes*, *recN* for *Cronobacter sakazakii*, *ipaH* for *Shigella flexneri*, *CarA* for *Pseudomonas putida*, *rfbE* for *Escherichia coli*, *vvhA* for *Vibrio vulnificus*, and *gyrB* for *Vibrio alginolyticus*. The sensitivity of the single system in this study was found to be 20, 1.5, 15, 15, 13, 14, 17, and 1.8 pg for *V. parahaemolyticus*, *L. monocytogenes*, *E. coli* O157:H7, *C. sakazakii*, *S. flexneri*, *P. putida*, *V. vulnificus*, and *V. alginolyticus*, respectively. The minimum detection limit of the multiplex system reaches pg/μL detection level; in addition, the multiplex system exhibited good specificity and stability. Finally, the assays maintained good specificity and sensitivity of 10^4^ CFU/mL for most of the samples and we used 176 samples of eight aquatic foods, which were artificially contaminated to simulate the detection of real samples. In conclusion, the multiplex PCR method is stable, specific, sensitive, and time-efficient. Moreover, the method is well suited for contamination detection in these eight aquatic foods and can rapidly detect pathogenic microorganisms.

## 1. Introduction

Foodborne pathogenic bacteria are biological agents that cause over 70% of foodborne illnesses. These bacteria can be transmitted through contaminated daily food and infant food, and an estimated 600 million people are infected worldwide each year. Thus, the development of a rapid, accurate, and sensitive means for detecting and identifying foodborne pathogenic bacteria is crucial in preventing the spread of foodborne diseases and ensuring food safety.

*Vibrio parahaemolyticus*, *Listeria monocytogenes*, *Cronobacter sakazakii*, *Shigella flexneri*, *Pseudomonas putida*, *Escherichia coli*, *Vibrio vulnificus*, and *Vibrio alginolyticus* are common pathogenic microorganisms, and some foods that are not properly handled are susceptible to these pathogenic microorganisms. *V. parahaemolyticus* is a human pathogen widely distributed in the marine environment. The inadvertent ingestion of *V. parahaemolyticus*-contaminated seafood can cause acute gastroenteritis [1]. *L. monocytogenes* ATCC 19115 is widely distributed in live fish in freshwater and coastal areas. Due to the increased consumption of food-related listeriosis, this strain has become a pathogenic microorganism of widespread concern [2]. Inadvertent infection can cause sepsis, meningitis, and other diseases. *E. coli* O157:H7 is a pathogenic microorganism that is widely distributed in water and soil. Cases of food poisoning and even epidemics caused by the microorganism have been reported in many countries and regions. *E. coli* O157:H7 can easily spread among ruminants and cause severe gastrointestinal diseases and hemolytic uremic syndrome [3]. *C. sakazakii* has been identified in drinking water and in a variety of foods, such as cheese and cured meats [4], and the inadvertent ingestion of contaminated food and water may cause bacteremia, meningitis, and other diseases. *S. flexneri* can cause infectious bacterial dysentery, which is exacerbated by a lack of clean water and poor sanitation in developing countries, causing multiple deaths [5]. *P. putida* is found in most aerobic temperate soil and water habitats. This microorganism is an opportunistic pathogen that can cause sepsis and urinary tract infections in immunocompromised populaces, and it is associated with host eye infections and diarrhea symptoms [6]. *V. vulnificus* is an opportunistic pathogen that can be isolated from water sediments and various seafood. This pathogenic microorganism can cause severe fulminant systemic infections with symptoms including fever, nausea, and shock when accidentally consuming its contaminated food, which is highly poisonous [7]. *V. alginolyticus* is a saline-loving bacterium found in marine and estuarine environments; it can infect the populace through contact with water or consumption of contaminated seafood. Several studies have reported the number of patients infected from 1988 to 2012. The reports showed that most infected patients were from coastal states and cities [8]. Generally, varieties of pathogenic microorganism groups coexist in contaminated food to cause various foodborne diseases. Therefore, the development of advanced technologies that can simultaneously detect and identify a variety of pathogenic microorganisms in a variety of foods is vital and urgent.

Because pathogenic bacteria can pose a greater risk to human health, there is a need to focus on their detection to reduce these bacteria in food. The traditional culture incubation method (gold standard) is generally used to detect bacteria. However, this method requires multiple experimental steps, including isolation and culture and biochemical and serological experiments. In addition, the operation of the traditional method is cumbersome and time-consuming and can only achieve a single test. As a result, the traditional culture method is ineffective in detecting the increasing number of bacterial species. Therefore, new detection methods and technologies need to be developed. Molecular biology detection technology, particularly the multiplex polymerase chain reaction (PCR) variant of conventional PCR, has been initially applied to simultaneously detect various pathogenic microorganisms. The method possesses the advantages of high efficiency, low cost, high speed, and high operability. The multiplex PCR assay is a technique recently developed to diagnose and identify multiple genomes simultaneously within four hours [9]. Multiplex PCR can be used to simultaneously detect highly virulent *L. monocytogenes* as the target strain at a minimum concentration of 10^4^ CFU/mL within 48 h [10]. Hernandez et al. successfully detected 1~10 CFU of *Shigella* and *Salmonella* from food samples within 24 h at low detection limits [11]. Li et al. also used multiplex PCR to detect the genomic DNA of four pathogenic bacteria with a detection limit of 100 pg/μL [12]. Ashraf et al. successfully detected pathogenic bacteria from milk samples at a concentration of 10^4^ CFU/mL [13]. The identification efficiency of multiplex PCR is higher than that of single PCR. In addition to reducing the detection time, multiplex PCR reduces the detection cost. Thus, multiplex PCR assays are important for the rapid and multiple identifications of pathogenic bacteria in mixed samples.

Although multiplex PCR assays have been widely used in various testing fields, no studies have reported the simultaneous detection of *V. parahaemolyticus*, *L. monocytogenes*, *C. sakazakii*, *S. flexneri*, *P. putida*, *E. coli* O157:H7, *V. vulnificus*, and *V. alginolyticus*. These eight foodborne pathogenic bacteria are widely present in aquatic food. Especially after improper storage or placement of aquatic products for a certain period, these pathogenic bacteria will grow wildly; some bacteria even produce toxins to cause serious harm to humans. Thus, adequate research is needed in this area. If single-tube amplification can be achieved, it can greatly reduce the cost of detection and determine the presence of eight pathogenic microorganisms efficiently and rapidly. In this study, we designed specific primers to detect eight pathogenic microorganisms in seafood simultaneously. This method is more efficient for detecting common pathogenic microorganisms in seafood and preventing foodborne illness.

## 2. Materials and Methods

### 2.1. Bacterial Strains and Culture Conditions

All strains used in this study are listed in Table 1. *V. parahemolyticus* was grown in tryptic soy broth (TSB; HKM, Guangzhou, China) medium with 3% (*w*/*w*) NaCl at 37 °C in a rotary shaker (Bluepard, Shanghai, China) at 180 rpm for 18 h. *L. monocytogenes*, *C. sakazakii*, and *S. flexneri* were grown in TSB medium at 37 °C in a rotary shaker at 180 rpm for 18 h. *P. putida* was grown in TSB medium at 30 °C in a rotary shaker, at 180 rpm for 18 h. *E. coli* O157: H7 was grown in Luria Bertani (LB; HKM, Guangzhou, China) medium at 37 °C in a rotary shaker at 180 rpm for 18 h. *V. vulnificus* and *V. alginolyticus* were grown in TSB medium with 3% (*w*/*w*) NaCl at 30 °C in a rotary shaker at 180 rpm for 18 h.

### 2.2. Primers

The design of multiplex PCR primers is often more complicated than that of single PCR primers. The addition of more primer pairs can result in the explosive growth of primer dimers and mismatches [14]. In addition, the multiplex PCR primers can analyze the primer dimer and mismatch generation using bioinformatics means, achieving optimal primer synthesis by modifying the primer length, GC value, and ΔTm.

The specific primers in this study were designed based on the target genes of the eight pathogenic microorganisms: the *toxS* in *V. parahaemolyticus* was used as a target gene. According to a previously reported study, a *toxS*-targeted PCR method could be applied in epidemiological investigations [15]; For *L. monocytogenes*, *vir.R* was used as a target gene (SN/T 2754.4-2011, Guangzhou, China) for primer design [16,17]; *rfbE* was a commonly used specific gene of *E. coli* O157:H7 [18]. Several studies have used this segment of the target gene to achieve rapid detection [19]. The *recN* gene of *C. sakazakii* was used as a target gene that could directly determine the genetic similarity between species of *Enterobacteriaceae* [20]. The detection of *S. flexneri* tends to treat *ipaH* as the target gene to design primers that can achieve efficient detection [21]. Previous studies applied the *carA* gene in the development of species-specific primers and multiplex PCR assays for *P. putida* [22]. Numerous studies also used *CarA* virulence gene in *P. putida* to design primers and retrograde real-time PCR to detect *P. putida* in water [23], both of which could achieve high specificity and sensitivity. Moreover, *V. vulnificus* use *vvhA* as the target gene. Previous studies have established *vvhA* as the target gene for rapid and accurate detection of *V. vulnificus* [24]. The *gyrB* gene of *V. alginolyticus* was a target gene to design primers that can also achieve rapid and highly sensitive detection [25].

These primers were designed using Premier 5.0 and Oligo 6.0, and the sequence specificity was evaluated using the NCBI primer-BLAST tool. All the primers used in this study were synthesized by Shanghai Sangon Biotech Co., Ltd. The information on primers is shown in Table 2.

### 2.3. DNA Extraction and Concentration Measurement

The strains used in standard experiments were cultured under suitable conditions. The DNA was extracted from the bacteria using the bacterial genomic DNA extraction kit (Vazyme Biotech Co., Ltd., Nanjing, China). The concentration and purification of the DNA were examined using a Nanodrop 2000 UV spectrophotometer (Thermo Scientific, Wilmington, NC, USA). Then, the genomic DNA was stored at −20 °C.

The genomic DNA of the artificially contaminated sample was extracted according to our previously established method (CN 105400774 A). The genomic DNA was extracted from a 1000 μL sample solution, and the culture medium was removed by centrifuging the bacteria at 10,760× *g* for 1 min and then washed thrice with phosphate buffer solution (PBS). Ten microliters of Sodium dodecyl sulfate (SDS, 1%, *w*/*w*) were added to the bacterium precipitate and shaken in a vortex for 5 min. Then, the mixture was diluted with 490 μL of double-distilled water and stored at −20 °C.

### 2.4. Optimization of PCR Reaction Conditions

All primer pairs were tested in a single PCR at a suitable annealing temperature to verify that the product bands appeared in the correct positions for effective validation of the amplification temperature before the commencement of the multiplex experiments. The first experiment optimized the conditions of single PCR according to the primer melting temperature (Tm). In this study, annealing temperature gradients were set from 48 °C to 55 °C at one °C per step. The annealing temperatures were determined using electrophoresis results according to the strip brightness (analyzed and calculated by Image Lab software). Subsequently, annealing temperature gradients were set from 48 °C to 58 °C for the multiplex PCR reaction. Finally, the suitable annealing temperature of the PCR system was optimized from two results.

In this study, the multiplex PCR mixture contained 25 μL of the 2 × GS Taq PCR Mix (Beijing Genes and Biotech Co., Ltd., Beijing, China), 2 μL of different primers (10 μM), and 1 μL of the target DNA templates. The mixture volume was filled to 50 μL with double-distilled water. Amplification was performed using the PCR Thermal Cycler Dice TP600 (Takara, Kusatsu, Shiga, Japan) under the conditions shown in Table 3.

### 2.5. Sensitivity of Single PCR

To verify the sensitivity of the PCR, the template was diluted in a gradient. Then, the sensitivity was assessed by the change in band brightness. The initial concentration of genomic DNA was diluted to 200 ng/μL for *V. parahaemolyticus*, 150 ng/μL for *L. monocytogenes*, *E. coli* O157:H7, and *C. sakazakii*, 130 ng/μL for *S. flexneri*, 140 ng/μL for *P. putida*, 170 ng/μL for *V. vulnificus*, and 180 ng/μL for *V. alginolyticus*. Then the concentration of genomic DNA was 10 times further diluted with sterilized double-distilled water to 10^−7^ times the initial concentration and amplified using the optimized PCR reaction system. Afterward, the limit of detection (LOD) of single PCR was determined using optimized procedures, verified by gel electrophoresis, and observed with GelDoc XR+ Imagelab (Bio-Rad Laboratories, Inc., Hercules, CA, USA).

In addition, the target strains were cultured in the same way as in Section 2.1. After two generations of strain activation, the strains were incubated for 18 h with the corresponding medium and culture temperature and then diluted to 10^−9^ with saline in a 10-fold gradient. The strains were cultured using the coated plate method [26] for colony observation and counting.

### 2.6. Optimization of Multiplex Primer Ratios and Detection of Sensitivity of Multiplex PCR

To understand the brightness of different bands and assess the specific amplification efficiency of different strains, the band brightness was finally balanced by modifying different primer ratios, increasing the primer concentration for low brightness bands and decreasing the primer concentration for high brightness bands. This step was performed by adjusting the concentrations and ratios of different primers. The specific primers added from group 1 to group 7 are shown in Table 4.

The sensitivity was tested after the determination of optimal reaction conditions for multiplex PCR. The total volume of mixed DNA in the multiplex system was 8 μL, which included 1270 ng of DNA. The original concentration of DNA template to 10^−6^ was diluted using the same practice as for sensitivity of single PCR and amplified using the optimized multiplex PCR mixture and reaction procedure, with subsequent determination of the LOD for multiplex reactions.

### 2.7. Specificity of Multiplex PCR and Stability of Multiplex PCR

This part of the experiment was achieved by detecting mixed primers and different DNA template combinations for amplification. DNA templates were extracted from common foodborne pathogenic bacteria, including *E. coli*, *Pseudomonas aeruginosa*, *Bacillus cereus*, *Yersinia enterocolitica*, *Streptococcus pyogenes*, *Salmonella enterica*, *Vibrio cholerae* (Vbo), *Vibrio minima*, *Staphylococcus aureus*, and *Klebsiella pneumoniae*. After the extraction, the extracts were amplified with an optimized multiplex PCR reaction system, which enabled us to obtain the primer specificity.

The stability experiments of multiplex PCR were performed by amplifying other groups of mixed-DNA templates. The optimized multiplex PCR program was used to amplify eight groups containing seven target DNA templates and eight groups containing only one target DNA template. The stability of the multiplex PCR was determined by whether the bands appeared in the correct positions in the electrophoresis results.

### 2.8. Evaluation of Artificially Contaminated Samples

The established multiplex PCR assay was applied to detect pathogenic microorganisms in the artificial samples to simulate the actual situation. Twenty-two samples were tested for each food: fresh basa catfish, beltfish, river shrimps, sea shrimps, scallops, oysters, seaweed, and skunk cabbage. The above food samples were purchased at the local supermarket (Guangzhou, China), transported to our laboratory, and operated on the ultra-clean table. First, the food samples were sterilized with 75% ethanol, washed with sterilized water, and then analyzed via the standard culture method (GB 4789.36-2016, China) to ensure the samples did not contain the eight target strains. The procedures were as follows: 25 g of samples were homogenized in 225 mL of sterilized phosphate buffer (a mixture of 8 g NaCl, 0.2 g KCl, 3.63 g Na_2_HPO_4_, and 0.24 g K_2_HPO_4_ diluted in 1 L of distilled water, PH 7.4) to generate 1:10 sample homogenate; 10^8^ CFU/mL of the eight target strains and 13 non-target strains were added to the sample homogenate to form a suspension with 10^7^ CFU/g [27,28]; and finally, the DNA template of sample homogenate was extracted using the second method in Section 2.3.

In addition, the sensitivity verification experiments for eight artificially contaminated target strains were conducted. Precisely, 10^8^ CFU/mL of various bacterial suspensions were added to the sample solution to prepare 10^7^ CFU/mL. A 10-fold gradient was diluted to 10^2^ CFU/mL with the sample solution. Then, the sample DNA was extracted as a template for verification according to the second method in 2.3.

## 3. Results

### 3.1. Optimization of PCR Annealing Temperature

The results of single PCR annealing temperature optimization showed that the specific primers amplification could be observed for *P. putida* at 500 bp (Figure 1A), *E. coli* O157:H7 at 110 bp (Figure 1B), *V. vulnificus* at 122 bp (Figure 1C), *L. monocytogenes* at 162 bp (Figure 1D), *C. sakazakii* at 186 bp (Figure 1E), *S. flexneri* at 140 bp (Figure 1F), *V. parahaemolyticus* at 89 bp (Figure 1G), and *V. alginolyticus* at 228 bp (Figure 1H). In addition, the brightness of the bands did not change significantly at the annealing temperature of 48~55 °C, indicating that the annealing temperature has no significant effect on single PCR.

Subsequently, the results of multiplex PCR annealing temperature optimization showed that the brightness of the bands significantly varied from 48~58 °C (Figure 2). To avoid the decrease in amplification specificity due to low annealing temperature and the darkening of small molecules due to high annealing temperature, 55 °C was finally chosen as the optimal annealing temperature for the subsequent experiments.

### 3.2. Single PCR Assay Sensitivity Validation and Traditional Method

The sensitivity of single PCR was verified by diluting the genome concentration, and the results showed that the method effectively achieved a low DNA concentration detection limit. The genomes of *P. putida* with only 14 pg (Figure 3A), *V. alginolyticus* with 1.8 pg (Figure 3B), *V. vulnificus* with 170 pg (Figure 3C), *V. parahaemolyticus* with 20 pg (Figure 3D), *C. sakazakii* with 15 pg (Figure 3E), *S. flexneri* with 13 pg (Figure 3F), *E. coli* O157:H7 with 15 pg (Figure 3G), and *L. monocytogenes* with 1.5 pg (Figure 3H) of DNA were successfully detected.

Compared with the number of countable colonies obtained from the experiments using a traditional method, the results of this experiment using the PCR assay exhibited excellent detection ability. In addition, the PCR assay can effectively obtain the test results within three h, while the traditional assay can take about 48 h. Moreover, multiplex PCR is characterized by multiple indicators, short time, good sensitivity, and low cost.

### 3.3. Optimization of Multiplex PCR Primer Ratios and Sensitivity

The brightness of the multiplexed bands was adjusted by adjusting the primer concentrations and ratios in the multiplexed system, and the results are shown in Figure 4. The brightness of this band (186 bp) was significantly reduced by reducing the addition of *C. sakazakii* specific primers in lanes 2~7, and the brightness of the corresponding band of *L. monocytogenes* (162 bp) was greater. The individual bands of lane 4 were bright and well-defined. Thus, we chose a primer ratio of 1:2.5:1:0.5:0.5:1:1:0.5 (for *V. parahaemolyticus*, *L. monocytogenes*, *E. coli* O157:H7, *C. sakazakii*, *S. flexneri*, *P. putida*, *V. vulnificus*, and *V. alginolyticus*) for the subsequent experiments.

The sensitivity of the multiplex PCR was also verified by mixing the DNA of the target strain (1~10^−6^ times the original concentration) with sterilized double-distilled water as the template of the system. The results of the multiplex PCR assay showed that it could properly identify the presence of bacteria at 200, 15, 150, 15, 130, 14, 170, and 180 pg/µL of DNA template for *V. parahaemolyticus*, *L. monocytogenes*, *E. coli* O157:H7, *C. sakazakii*, *S. flexneri*, *P. putida*, *V. vulnificus*, and *V. alginolyticus*, respectively (Figure 5).

### 3.4. Results of Multiplex PCR Specificity and Stability

The results of the multiplex PCR specificity experiments showed that the multiplex PCR system could accurately distinguish between mixed DNA and single DNA. In Lane 1 of Figure 6, the genes of the eight target strains used as templates and the control could successfully amplify eight bands. Lanes 2 to 9, used as single-target DNA templates, showed that the positions of the amplified bands corresponded to each of the eight specific strains. Lanes 10 to 23 were single non-target DNA templates with no amplified bands appearing, and lane 24 was a negative control without any template for amplification. It is worth noting that the amplified band appeared at about 400 bp in lane 9; however, the position was not within eight target lengths, and the band disappeared after amplifying mixed templates without affecting the experimental results. In summary, no bands were observed for non-target bacteria, and correct bands for target bacteria indicated that the amplification system showed good specificity.

The stability results test showed that specific target bands were found in all positive samples, while no amplification products were found in the negative samples (Figure 7). The above results were in accordance with our expectations and experimental requirements. That is, these results showed that the established multiplex PCR method has good stability and reproducibility.

### 3.5. Artificial Contamination Sample Validation

The optimized multiplex PCR was applied to detect artificially contaminated basa catfish, beltfish, river shrimps, sea shrimps, scallops, oysters, seaweed, and skunk cabbage samples. Except for the negative control without template addition, all food samples were randomly contaminated with each of the 21 strains used in this paper, and each sample was verified with 22 contamination groups. The DNA of the contaminated samples was subsequently extracted as a template for validation, and the bands were obtained. The electrophoresis results of basa fish and beltfish are shown in Figure 8A, the results of river shrimps and sea shrimps are shown in Figure 8B, the results of scallops and oysters are shown in Figure 8C, and the results of seaweed and skunk cabbage are shown in Figure 8D. All these results showed that this multiplex assay exhibited good specificity in detecting artificially contaminated samples; therefore, it can be applied to detect real samples.

Except for the sample skunk cabbage, the sensitivity test results showed that the artificially contaminated sample detection limit was 10^4^ CFU/mL, and the above results are presented in Table 5. Moreover, the overall detection rate was 100% after the statistical analysis.

## 4. Discussion

This study designed a multiplex PCR method that can simultaneously detect eight pathogenic microorganisms and reduce the detection cost to achieve high efficiency in detecting pathogenic microorganisms in food. The eight pathogenic microorganisms studied in this paper are common in daily food, aquatic products, and infant food. These pathogenic microorganisms also produce pathogenic toxins, a significant cause of foodborne illness. So, both the pathogenic microorganisms and the toxins they have can threaten human health. These hazards include but are not limited to food poisoning, intestinal infections, zoonotic infections, and parasitic diseases. Therefore, the detection of pathogenic microorganisms in these products is needed to prevent foodborne diseases and reduce their transmission.

For single reactions, all the different annealing temperatures can be amplified; for multiplex reactions, lowering the appropriate annealing temperature is more beneficial to the amplification of all target genes [29]. Thus, a low annealing temperature of 55 °C was selected as the optimal temperature to avoid primer mismatches (caused by too low annealing temperature) for subsequent reaction annealing temperature.

In this study, the amplification efficiency of *C. sakazakii* was significantly higher than the rest of the strains under the same reaction conditions. In contrast, the amplification efficiency of *L. monocytogenes* had the darkest bands. Thus, the amplification efficiency was improved by adjusting the primer concentration [30] to effectively balance the amplification efficiency of each primer pair and the brightness of the bands. This study showed that the amplification efficiency of *C. sakazakii* was higher under the same amplification conditions, decreasing the amplification efficiency of the remaining fragments and reducing band brightness. The above strategy adjusted the primer ratios, and the overall brightness and integrity of the eight bands from the electrophoresis results were evaluated comprehensively (calculated by analysis with Image Lab software). The bands obtained by this screening method have high brightness and good homogeneity, which can avoid errors caused by visual observation.

From current sensitivity results, the LOD of *E. coli* in our study was similar to the results previously reported [12], with a detection limit of 10 pg. Nogva et al. performed PCR amplification using the *HlyA* gene of *L. monocytogenes*, and the results showed a detection limit of 1 pg of *L. monocytogenes* [31], similar to the results of the current experiment. Another study [32] reported the detection limits of 200 pg for *V. parahaemolyticus*, which were one order of magnitude lower than those in this study. Wang et al. reported a detection limit of about 10^3^ CFU for *P. putida* [33], similar to this study. Duplex PCR was used to detect *V. vulnificus* with a detection limit of 100 pg [34], while the detection limit of 17 pg for *V. vulnificus* was obtained in this study, which was an order of magnitude lower compared to that of the article. Sun et al. [35] detected *C. sakazakii* using a PCR with a detection limit of 10 pg, similar to the detection limit in this paper. Suvash et al. detected *S. flexneri* using real-time PCR with a detection limit of 100 pg [36], while the LOD for *S. flexneri* in the present study was 13 pg. Zhao et al. showed that the detection limit for *V. alginolyticus* using PCR was 10 pg [37], which was higher than the results of the present study. Among the results of multiplex amplification, the detection limits of *P. putida* and *C. sakazakii* were comparable with those of the current research for single-duplex amplification and more sensitive than those of the previous article. The detection limits of *V. parahaemolyticus*, *S. flexneri*, and *V. vulnificus* were similar to those in the previous article for single-duplex amplification. Amplification reached pg/μL levels for either target strain, indicating that the multiplex PCR assay in this study exhibited high sensitivity and achieved a lower detection limit.

This method effectively amplified mixed and individual samples but failed to amplify non-target strains mixed in those samples. The results show that the method is more specific and can be used to detect and identify specific pathogenic bacteria.

## 5. Conclusions

In this study, we established an economical and convenient multiplex PCR assay technique to simultaneously detect *V. parahaemolyticus*, *L. monocytogenes*, *C. sakazakii*, *S. flexneri*, *P. putida*, *E. coli*, *V. vulnificus*, and *V. alginolyticus* contamination in seafood. The method is suitable for analyzing and detecting pathogenic microorganisms in various seafood products. The reaction system configuration and amplification can be completed in 90 min. In addition to seafood, this method can be applied to other food samples to achieve the low-cost, high-efficiency, rapid, specific, and sensitive detection of food pathogenic bacteria. Furthermore, the multiplex PCR assay is more efficient and less costly than the single PCR assay. Thus, this method can be widely used to assess food quality and provide technical support for food safety.

## Figures and Tables

**Figure 1 foods-11-03909-f001:**

Single PCR amplification at the different annealing temperatures. Lane M: DNA marker B 600; Lane 1~8: Annealing temperature is 48 °C, 49 °C, 50 °C, 51 °C, 52 °C, 53 °C, 54 °C, and 55 °C, respectively. (**A**) *P. putida*; (**B**) *E. coli* O157:H7; (**C**) *V. vulnificus*; (**D**) *L. monocytogenes*; (**E**) *C. sakazakii*; (**F**) *S. flexneri*; (**G**) *V. parahaemolyticus*; (**H**) *V. alginolyticus*.

**Figure 2 foods-11-03909-f002:**
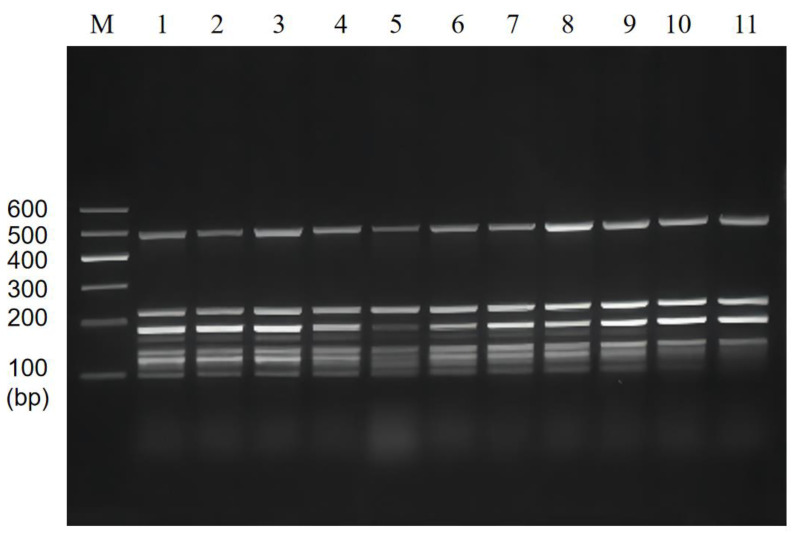
Multiplex PCR amplification at the different annealing temperatures. Lane M: DNA marker B 600; Lane 1~11: Annealing temperature is 48 °C, 49 °C, 50 °C, 51 °C, 52 °C, 53 °C, 54 °C, 55 °C, 56 °C, 57 °C, and 58 °C, respectively.

**Figure 3 foods-11-03909-f003:**
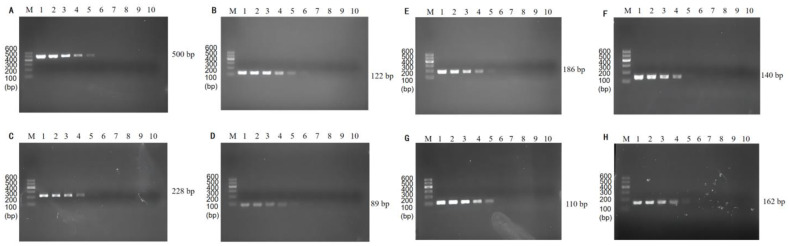
Results of single PCR sensitivity. Lane M: DNA marker B 600; Lane 1~8: Dilute the DNA template to the original concentration of 1, 10^−1^, 10^−2^, 10^−3^, 10^−4^, 10^−5^, 10^−6^, 10^−7^, 10^−8^, and 10^−9^ times, respectively. (**A**) The initial concentration of the template for *P. putida* is 140 ng/μL; (**B**) The initial concentration of the template for *V. alginolyticus* is 180 ng/μL; (**C**) The initial concentration of the template for *V. vulnificus* is 170 ng/μL; (**D**) The initial concentration of the template for *V. parahaemolyticus* is 200 ng/μL; (**E**) The initial concentration of the template for *C. sakazakii* is 150 ng/μL; (**F**) The initial concentration of the template for *S. flexneri* is 130 ng/μL; (**G**) The initial concentration of the template for *E. coli* O157:H7 is 150 ng/μL; (**H**) The initial concentration of the template for *L. monocytogenes* is 150 ng/μL.

**Figure 4 foods-11-03909-f004:**
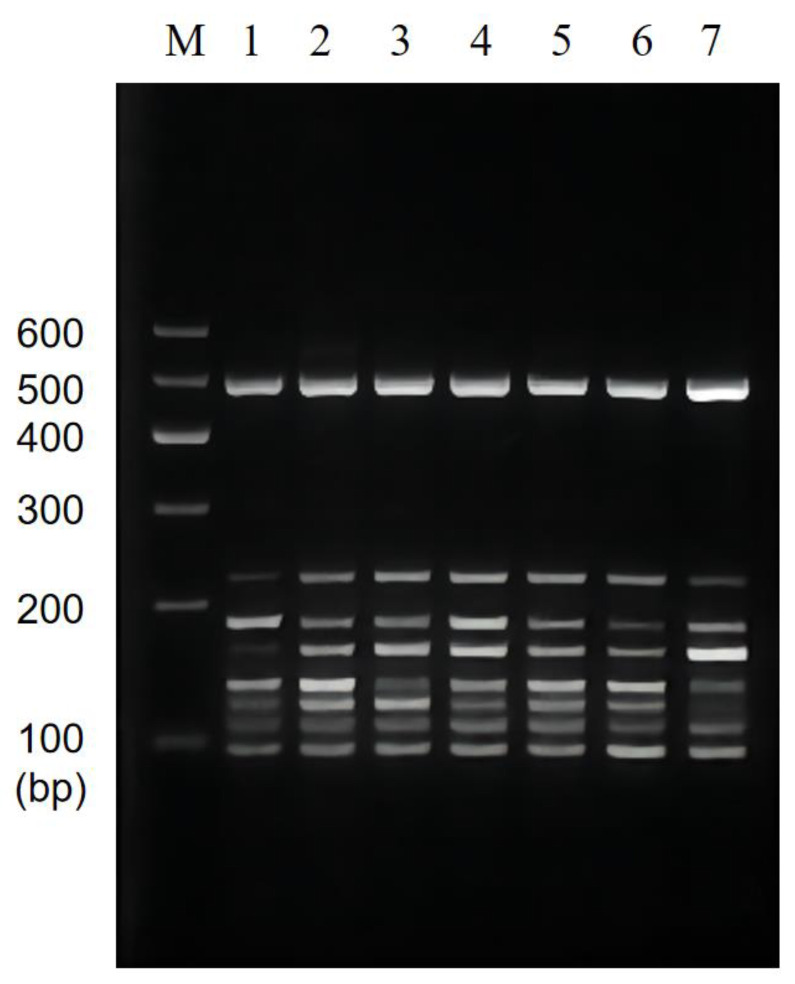
Results of primer ratios optimization for multiplex PCR. Lane M: DNA marker B 600; Lane 1~7: The amplification results of different primer concentration ratios. The lightest band is Lane 4, the ratio of primer of *V. parahaemolyticus*, *L. monocytogenes*, *E. coli* O157:H7, *C. sakazakii*, *S. flexneri*, *P. putida*, *V. vulnificus*, and *V. alginolyticus* is 1:2.5:1:0.5:0.5:1:1:0.5. The proportion of primers used for the rest is shown in Table 4.

**Figure 5 foods-11-03909-f005:**
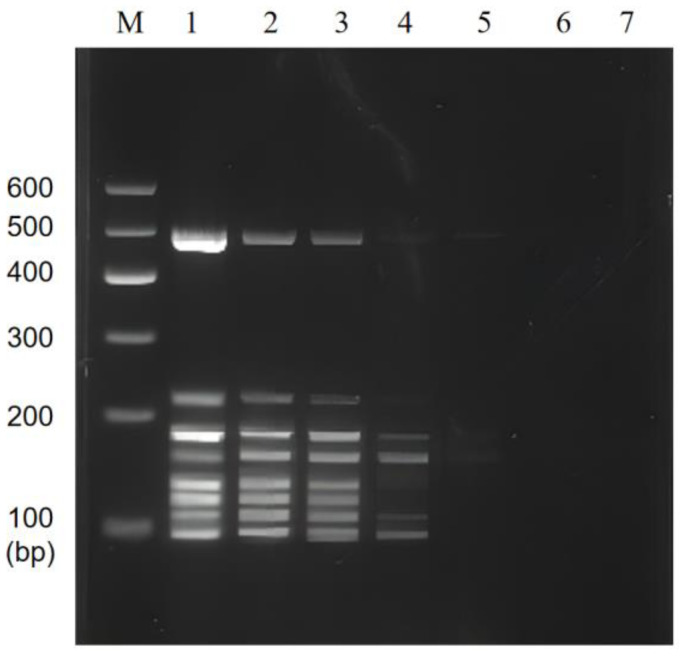
Results of multiplex PCR sensitivity. Lane M: DNA marker B 600; Lane 1~7: Mixed DNA was 1~10^−6^ times the original concentration.

**Figure 6 foods-11-03909-f006:**
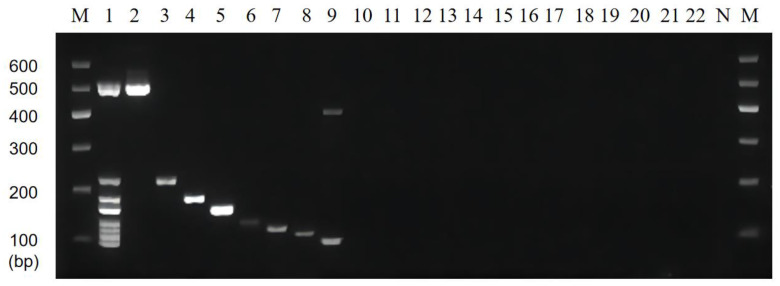
Results of multiplex PCR specificity. Lane M: DNA marker B 600; Lane 1: Mixed 8 target DNA as templates; Lane 2~22: The DNA templates were *P. putida*, *V. vulnificus*, *C. sakazakii*, *L. monocytogenes*, *S. flexneri*, *V. alginolyticus*, *E. coli* O157:H7, *V. parahaemolyticus*, *E. coli* EAEC, *E. coli* ETEC, *S. pyogenes*, *Y. enterocolitica*-1, *Y. enterocolitica*-2, *S. aureus*, *P. aeruginosa*, *V. minima*, *V. cholerae* (Vbo), *B. cereus*, *E. coli* EPEC, *S. enterica*, and *K. pneumoniae*, respectively; Lane N: Negative control group without the addition of DNA template.

**Figure 7 foods-11-03909-f007:**
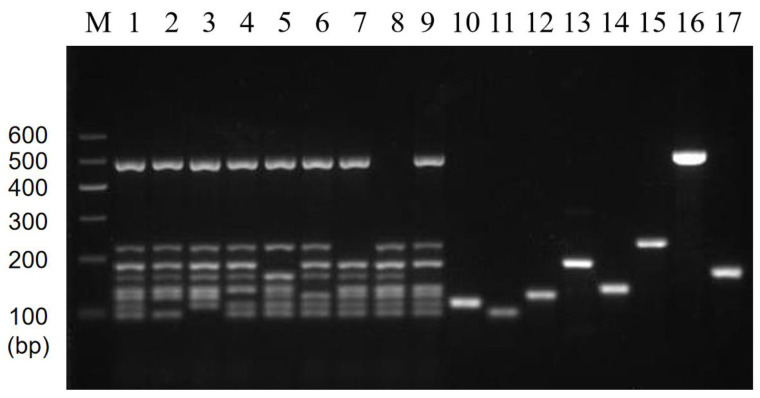
Results of multiplex PCR stability. Lane M: DNA marker B 600; Lane 1: Mixed 8 target DNA as templates; Lane 2~9: 8 mixed templates based on 7 templates missing *E. coli* O157:H7, *V. parahaemolyticus*, *V. alginolyticus*, *C. sakazakii*, *S. flexneri*, *V. vulnificus*, *P. putida*, and *L. monocytogenes*, respectively; Lane 10~17: Single DNA templates *E. coli* O157:H7, *V. parahaemolyticus*, *V. alginolyticus*, *C. sakazakii*, *S. flexneri*, *V. vulnificus*, *P. putida*, and *L. monocytogenes*, respectively.

**Figure 8 foods-11-03909-f008:**
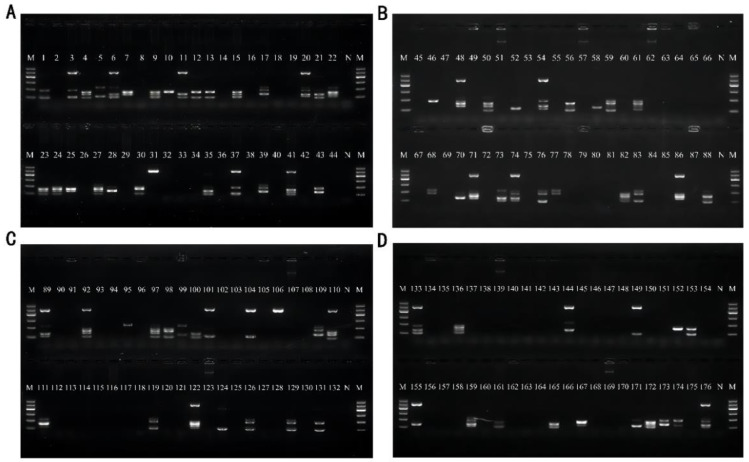
Results of artificial contamination of samples. Lane M: DNA marker B 600; Lane N: Negative control group without the addition of DNA template for amplification. (**A**) Sample basa catfish for Lane 1–22 and sample beltfish for Lane 23~44; (**B**) Sample River shrimp for Lane 45~66 and Sample Sea shrimp for Lane67~88; (**C**) Sample scallops for Lane 89~110 and Sample oysters for Lane 111~132; (**D**) Sample seaweed for Lane 133~154 and Sample skunk cabbage for Lane 155~176.

**Table 1 foods-11-03909-t001:** Bacteria strains were used in this study.

Number	Bacterial Species	Source
Target Strains		
1	*Vibrio parahaemolyticus*	ATCC 17802
2	*Listeria monocytogenes*	ATCC 19115
3	*Cronobacter sakazakii*	ATCC 35150
4	*Shigella flexneri*	ATCC 12022
5	*Pseudomonas putida*	SCAUFC-1
6	*Escherichia coli* O157: H7	ATCC 29544
7	*Vibrio vulnificus*	ATCC 27562
8	*Vibrio alginolyticus*	ATCC 33707
**Non-Target Strains**		
9	*Escherichia coli* EPEC	CICC 10412
10	*Escherichia coli* EAEC	CICC 24186
11	*Escherichia coli* ETEC	CICC 10667
12	*Pseudomonas aeruginosa*	CMCC(B) 10104
13	*Bacillus cereus*	CMCC(B) 63303
14	*Yersinia enterocolitica-1*	CICC 10869
15	*Yersinia enterocolitica-2*	CMCC(B) 52204
16	*Streptococcus pyogenes*	IQCC 22107
17	*Salmonella enterica*	ATCC 14028
18	*Vibrio cholerae* (Vbo)	FSCC 232004
19	*Vibrio minima*	ATCC 33653
20	*Staphylococcus aureus*	CMCC(B)26003
21	*Klebsiella pneumoniae*	SCAUFC-2

ATCC, American Type Culture Collection, USA; CICC, China Center of Industrial Culture Collection; CMCC, National Center for Medical Culture Collections; IQCC, Chinese Academy of Inspection and Quarantine Culture Collections; SCAUFHSM, Food College of South China Agricultural University.

**Table 2 foods-11-03909-t002:** Primers were used in this study.

Bacteria	Target Genes	Primer Sequences (5′-3′)	Product Size (bp)
*V. parahaemolyticus*	*toxS*	F-TTTTGGCCGTATCTATCCTT	89
		R-CTGCCATTCATTTGATGTAAGC	
*L. monocytogenes*	*vir.R*	F-GTCTAGTAGAAAAGAAGAAGTCC	162
		R-GGTTTCCCAGGAAGTTTG	
*E. coli* O157:H7	*rfbE*	F-AAAACACTTTATGACCGTTGT	110
		R-GCGCAGATATTTGTCATCCT	
*C. sakazakii*	*recN*	F-CATATGGGTTTCGGTCATCGC	186
		R-GCTGATTTTCGATGAAGTGGACG	
*S. flexneri*	*ipaH*	F-GAAAGCCTACCAGCCGTA	140
		R-TCTTCGAGGATGATAGTGC	
*P. putida*	*CarA*	F-AGGAAATCCTTACAGACCCT	500
		R-CAGGATGTTCAGCTTGACG	
*V. vulnificus*	*vvhA*	F-TCCGATCGTTGTTTGACCGTA	228
		R-TTTGACTTGTTGTAATGTGGGTT	
*V. alginolyticus*	*gyrB*	F-CATTCCTGAACTCTGGTGT	122
		R-CGTTTTGTTGGTGTTTAGGT	

**Table 3 foods-11-03909-t003:** Optimization of amplification conditions.

Optimization of Amplification Conditions			
Single-Duplex PCR Amplification			
Steps	Temperature	Time	Number of Cycles
Initial denaturation	95 °C	3 min	1
Denaturation	94 °C	25 s	30
Annealing	48–55 °C	25 s	
Initial extension	72 °C	10 s	
Final extension	72 °C	5 min	1
**Multiplex PCR Amplification**			
**Steps**	**Temperature**	**Time**	**Number of Cycles**
Initial denaturation	95 °C	3 min	1
Denaturation	94 °C	25 s	25
Annealing	48–58 °C	25 s	
Initial extension	72 °C	10 s	
Final extension	72 °C	5 min	1

**Table 4 foods-11-03909-t004:** Seven groups of primer ratio.

Group	Primer Addition (μL)							
	*toxS*	*vir.R*	*rfbE*	*recN*	*ipaH*	*CarA*	*vvhA*	*gyrB*
**1**	1	1	1	1	1	1	1	1
**2**	1	1.5	1	0.5	1	1	1	1
**3**	1	2	1	0.5	0.5	1	1	1
**4**	1	2.5	1	0.5	0.5	1	1	0.5
**5**	1	1	1.5	0.5	1	1	1	1
**6**	1.5	2	1	0.5	1	1	1	1
**7**	1	3	1	0.5	0.5	1	0.5	0.5

**Table 5 foods-11-03909-t005:** Contamination of samples.

Contaminated Samples	Situation of Artificial Pollution (Positive/Total)	Multiplex PCR Assay (Positive/Total)	Detection Rate (%)	Detection Sensitivity (CFU/mL)
Basa catfish	16/22	16/22	100	10^4^
Beltfish	12/22	12/22	100	10^4^
River shrimps	9/22	9/22	100	10^4^
Sea shrimps	11/22	11/22	100	10^4^
Scallops	12/22	12/22	100	10^4^
Oysters	7/22	7/22	100	10^4^
Seaweed	6/22	6/22	100	10^4^
Skunk cabbage	10/22	10/22	100	10^5^

## Data Availability

Data are contained within the article and available upon reasonable request from the corresponding author.

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
