# Peer review of "The Detection of Foodborne Pathogenic Bacteria in Seafood Using a Multiplex Polymerase Chain Reaction System"

_foods, 2022, doi:10.3390/foods11233909_

Round 1

Reviewer 1 Report

In their manuscript "Detection of Foodborne Pathogenic Bacteria in Seafood using a Multiplex Polymerase Chain Reaction System" Li et al. described an attempt to develop a multiplex PCR procedure for the rapid and sensitive identification of 8 pathogenic bacteria occurring in seafood products. Although the method seems to work, there is a great number of previously published similar papers dealing with the same microorganisms (in different combinations) but much more elaborated, so I do not feel that this paper contributes significantly to the topic. Also, in my experience, the 89, 110, and 122 bp fragments are hardly visible to identify those pathogenic bacteria occurring in real food products. Thus, false negatives can occur when running a real test. Also, there are some typos, spelling, and grammar issues that should be corrected. Taken all together, my opinion is that the manuscript is not suitable for publication.

Author Response

We sincerely appreciate the time and effort and all the comments and suggestions by the reviewers to improve our manuscript. However, after carefully reading the reviewer’s comments, we would like to raise a few concerns regarding our manuscript.

First, we applied multiplex PCR assay to detect V. parahaemolyticus, L. monocytogenes, C. sakazakii, S. flexneri, P. putida, E. coli O157:H7, V. vulnificus, and V. alginolyticus. We optimized different parameters of both single and multiplex PCR systems, and the minimum detection limit of the multiplex system reached pg/μL detection level. The multiplex system exhibited good specificity and stability. The method is suitable for analyzing and detecting pathogenic microorganisms in various seafood products. The reaction system configuration and amplification time can be completed in 90 min. In addition to seafood, this method can be applied to other food samples to achieve low-cost and high-efficiency detention, providing a rapid, specific, and sensitive detection of food pathogenic bacteria. Although many studies have applied multiplex PCR to detect foodborne pathogens, this specific combination of 8 pathogenic microorganisms (including Gram-positive and Gram-negative bacteria) has not been analyzed. Meanwhile, few articles report the simultaneous detection of more than six different pathogenic bacteria, which is of interest to public health.

Second, another major concern is “the 89, 110, and 122 bp fragments are hardly visible to identify those pathogenic bacteria occurring in real food products. Thus, false negatives can occur when running a real test”. Regarding the blurred bands raised by the reviewer, the 89, 110, and 120 bp fragments were clearly visible in Figures 4, 5, and 7 in our text. They can be further distanced from similar bands by extending the electrophoresis time. For the case of false negatives, we did experiments for specificity and experiments for verification of artificially contaminated samples, and all of these experiments showed that no false negatives occurred.

Third, the manuscript was thoroughly edited by a professional manuscript editing service company with a large team of US-based certified language and scientific editors. We have also corrected and double-checked some typing and grammatical errors. Please see the details in our manuscript.

We would like to thank the reviewers again for your time and intellectual input to improve the quality of our manuscript.

Reviewer 2 Report

Attached please find the comments for the authors 

Author Response

We appreciate the reviewers' opinion that our article is worth publishing and has application prospects. We will also further improve the English expression of the paper and correct the mistakes or ambiguities. In addition, we also modified some unclear terms and the logic of the discussion part, hoping that readers can quickly obtain valuable information in the reading process.

We would like to thank the reviewers again for your time and intellectual input to improve the quality of our manuscript.

Reviewer 3 Report

Please improve according my suggested comments in the attached file.

Discussion section need to be improved. 

Author Response

We would like to thank the reviewers again for your time and intellectual input to improve the quality of our manuscript.

Reviewer 4 Report

This study has applied multiplex PCR assay for the detection of V. parahaemolyticus, L. monocytogenes, C. sakazakii, S. flexneri, P. putida, E. coli O157:H7, V. vulnificus, and V. alginolyticus. Although there are many studies with the application of multiplex PCR for detection of foodborne pathogens, this specific combination of pathogens has not been analyzed and it is of interest for the Public Health.

Some suggestions for improvement:

Line 28: please in this phrase: “These bacteria can be transmitted through contaminated food, and often infant an estimated 600 million people worldwide each year” add “infant food”

Line 33: please change all names of microorganisms to italics throughout the text

Line 41: Listeria monocytogenes is not a strain, it is a species, you must mention ATCC 19115 if you want to refer to the strain

Line 45: change “easxily

Line 57: please change “which can is highly poisonous”

Table 1: it would be useful to identify in this table the “target’ strains and the “non-target” strains used in the study

Line 177: “form seven ration groups” please change to “ratio”

Line 199-201: this phrase is confusing, please explain better

Line 390: “however, it failed to amplify other foodborne pathogenic or conditionally pathogenic bacteria causing diseases” , do you mean that the non-target bacteria have not been identified? If it is so, you should rephrase

Author Response

(The authors gave the same response as above.)

Round 2

Reviewer 1 Report

With the still limitions from the work, I am unable to accept for publication. 

Author Response

Thank the reviewers for their effective opinions. We have elaborated and supplemented the innovation and necessity of this work. This study is a combination that is not tested by other multiple PCR methods, which is very valuable for detecting seafood products. The following is a supplement to the innovation of the article portfolio.

“These eight foodborne pathogenic bacteria are widely present in aquatic food. Especially after improper storage or placement of aquatic products for a certain period, these pathogenic bacteria will grow wildly; some bacteria even produce toxins to cause serious harm to humans. Thus, adequate research is needed in this area. If single-tube amplification can be achieved, it can greatly reduce the cost of detection and determine the presence of eight pathogenic microorganisms efficiently and rapidly.”-L91-97